# Inter-Lineage Variation of Lassa Virus Glycoprotein Epitopes: A Challenge to Lassa Virus Vaccine Development

**DOI:** 10.3390/v12040386

**Published:** 2020-03-31

**Authors:** Francis Ifedayo Ibukun

**Affiliations:** Institute of Human Virology, University of Maryland School of Medicine, Baltimore, MD 21201, USA; FIbukun@ihv.umaryland.edu

**Keywords:** lassa virus, lassa virus diversity, glycoprotein, epitopes, lassa virus vaccine

## Abstract

Lassa virus (LASV), which causes considerable morbidity and mortality annually, has a high genetic diversity across West Africa. LASV glycoprotein (GP) expresses this diversity, but most LASV vaccine candidates utilize only the Lineage IV LASV Josiah strain GP antigen as an immunogen and homologous challenge with Lineage IV LASV. In addition to the sequence variation amongst the LASV lineages, these lineages are also distinguished in their presentations. Inter-lineage variations within previously mapped B-cell and T-cell LASV GP epitopes and the breadth of protection in LASV vaccine/challenge studies were examined critically. Multiple alignments of the GP primary sequence of strains from each LASV lineage showed that LASV GP has diverging degrees of amino acid conservation within known epitopes among LASV lineages. Conformational B-cell epitopes spanning different sites in GP subunits were less impacted by LASV diversity. LASV GP diversity should influence the approach used for LASV vaccine design. Expression of LASV GP on viral vectors, especially in its prefusion configuration, has shown potential for protective LASV vaccines that can overcome LASV diversity. Advanced vaccine candidates should demonstrate efficacy against all LASV lineages for evidence of a pan-LASV vaccine.

## 1. Introduction

Lassa fever (LF) is a zoonotic acute viral disease caused by Lassa virus (LASV), a member of the *Arenaviridae* family of RNA viruses. LASV is endemic to Nigeria and the Mano River Union (MRU) countries: Sierra Leone, Guinea, and Liberia in West Africa [1]. It has been suggested that the region of endemicity has broadened into countries bounded by these endemic areas, such as Benin, Togo, Ghana, Burkina Faso, Mali, and Cote D’Ivoire, in the past decade [2,3]. The population at risk of infection in West Africa is crudely projected to be 37.7 million and the annual incidence is estimated to be 300,000–500,000 cases with over 5000 deaths in endemic countries [4,5,6]. During the recent outbreak in Nigeria, about 810 confirmed cases were recorded with a case fatality rate (CFR) of 20.6% in 2019, which continued into 2020 with over 586 confirmed cases occurring from January to the second week of February [7,8].

LASV is mainly transmitted as a zoonosis, by ingestion of food and inhalation of aerosols contaminated with infected excreta of the multimammate rat species (*Mastomys natalensis*) which serve as a reservoir for LASV [9]. Broken skin exposure to infected rodent blood is also acknowledged as a risk factor for LF [10]. While LASV has been found in *M. erythroleucus* and *Hylomyscus pamfi* [11], there is no documented epidemiologic evidence of LASV transmission to humans from these species. Human-to-human transmission through infected body fluids such as “blood, feces, urine, throat swab, vomit, semen, and saliva”, occurs to a lesser extent but has caused important nosocomial outbreaks [4,12,13]. It is estimated that LASV infection is asymptomatic in 80% of the cases. In symptomatic incidents, LF is characterized by a flu-like illness with fever and a broad spectrum of symptoms such as malaise, headaches, sore throat, muscle aches, chest pain, cough, nausea, vomiting, diarrhea, and abdominal pain [1,4]. Persistent fever may herald severe cases with capillary bleeding from multiple organs and various patterns of organ failure [4]. LF is also associated with high maternal mortality rates in pregnancy and fetal loss [1,4]. Neurological sequelae, such as aseptic meningitis, and encephalopathy may occur in survivors and hearing loss is found in about 29% of patients [4].

LASV is a very diverse mammarenavirus. It has a core of two single-stranded RNA genome segments called Large (L) and Small (S) RNA segments which have an overall strain variation of 32% and 25% respectively, which is higher than the *Zaire ebolavirus* (EBOV) that is 97% conserved among its sequenced strains [12]. The L segment encodes the RNA-dependent RNA-polymerase and Z-protein genes, while the S segment encodes the glycoprotein (GP) and nucleoprotein (NP) genes. LASV has four confirmed lineages, I–IV, and three additional lineages discovered over the past decade based on phylogenetic analysis of human and rodent samples [11,12,14,15,16]. These lineages are distributed in geographically determined clusters, but not closely associated to host-species [12]. Lineages I–III are found Nigeria [12]: Lineage I comprises the first strain isolated in Lassa, North-Eastern Nigeria, while Lineage II has seven sub-lineages mainly distributed in Southern Nigeria, and Lineage III has five sub-lineages circulating around North-Central Nigeria (Figure 1) [17]. Lineage IV is found in Sierra Leone, Guinea, and Liberia [12] and is subdivided in two clades (IV.A and IV.B) in Liberia, with up to 25% genomic divergence from the Lassa Josiah strain widely utilized in vaccine and diagnostic assays [18]. The proposed additional lineages include: Lineage V from Mali and Cote D’Ivoire which is possibly a sub-lineage of Lineage IV, a possible sixth lineage comprising LASV “Kako” strains found in *H. pamfi* and related human strains from Nigeria, and the seventh proposed LASV lineage consists of strains from Togo [11,15,16]. The LASV strains of the proposed sixth lineage share a distant common ancestor with Lineage I on both the L and S RNA segment genes [11,16], whereas the Togo LASV strains’ RdRp genes (on the L RNA segment) clusters with Lineage II and its S RNA segment genes cluster with Lineage I. LASV strains from Benin and Ghana also clustered with the Togo LASV strains in a recent phylogenetic analysis [19].

LASV lineages also differ in pathogenicity in animal models and humans. Inbred strain 13 guinea pigs succumb to LASV infection from the Lineage IV Josiah strain and Lineage II strain [20,21,22], but not to Lineage I [23]. All outbred Hartley guinea pigs die from the LASV Lineage III strain infection [24], whereas approximately 30% of these guinea pigs develop lethal disease from wild type Lineage IV Josiah strain infection [20]. The Mali strain of the proposed LASV lineage V is less virulent in cynomolgus monkeys, causing atypical clinical features in these primates [25]. It diverges from the non-human primate (NHP) lethal disease phenotype of Lineage IV. Furthermore, predominant hepatocyte damage which is typified by low AST/ALT ratio and renal failure were common features in Nigerian human clinical cases [26], and LASV Lineage II NHP models also exhibit a similar clinical disease with predominant hepatic involvement [27]. Whereas, a high AST/ALT ratio may indicate predominant non-liver involvement in LF cases from Sierra Leone and Liberia [27]. Lineage IV LASV human infections also have more abundant viral genomes (a correlate of viral load) and a higher codon adaptation index compared to Nigerian lineages [12]. However, these have not been shown to cause the higher LF CFR seen in the MRU, however other factors such as delay in seeking care, economy, poor regional healthcare system, and host genetic factors also contribute to the regional difference in CFR [12]. The relatively high genomic diversity of LASV also impacts the molecular diagnosis of LF, especially nucleic acid amplification techniques. Mismatch between the primer or probe and the template sequence has led to primer redesign and/or multiple probe use for LASV RT-PCR techniques [28].

LASV diversity is reflected in its surface antigen, the glycoprotein (GP) (Figure 2), and to a larger extent in its nucleoprotein (NP) [14]. This GP is commonly utilized as immunogen in vaccine development, with the addition of NP in fewer candidates. LASV GP is translated as a 76-kDa precursor polyprotein (GPC). A host cell signal peptidase co-translationally cleaves off the stable signal peptide (SSP), then the GP precursor is N-glycosylated in the endoplasmic reticulum, and it is cleaved by a host protease (SKI-1-S1P) into GP1 and GP2 subunits [29,30,31]. The GP is embedded in the virion envelope surface as a trimeric complex comprised of monomers GP1, GP2, and SSP [32,33,34]. The GP1 and GP2 subunits of the GP trimeric complex interact in an intricate metastable manner, both existing in varying conformations relative to pH and receptor binding status [35]. Glycans are located at all 11 sites of N-glycosylation [30], with limited sections for receptor binding by the GP1, fusion by the GP2, and GP1/GP2 trimeric interactions remaining unshielded, thus rendering the protein less susceptible to antibody binding [32]. The metastable nature and extensive glycosylation is useful for humoral immunity evasion and viral fitness, complicating vaccine design efforts [36,37,38].

In order to reduce the LF burden, an effective LASV vaccine is needed in addition to other prevention, diagnostic and treatment strategies. There are advanced LASV candidates in development to fulfill this unmet need. However, most LASV vaccine candidates have utilized only the GP from the Lineage IV-Josiah strain as an immunogen and for many of the them, vaccinated animals were only challenged with the same LASV Josiah strain [27,42]. It is important to explore the implications of using a single lineage-LASV GP as a vaccine antigen in the face of LASV diversity. Evidence from human survivor, vaccine, and therapeutic monoclonal antibody (mAb) studies have shown that adaptive immune protection in LASV infection is probably conferred mainly by a cell-mediated immune response that is dependent on the early activation of innate immune and inflammatory pathways, especially for Type I IFN response [43,44,45,46,47,48,49]. While it is speculated that non-neutralizing antibody (non-NAb) dependent ADCC/ADCP functions is the likely purveyor of humoral immune protection, as seen in LASV vaccine studies in animal models [50,51]. There is no evidence for this role in human LF cases. Whereas, neutralizing antibodies (NAb) arise too late in natural human LASV infection to contribute to clinical recovery [36]. Moreover, the in-vitro neutralizing activity of anti-LASV human monoclonal antibodies does not correlate with protection in animal models [45,52]. In view of these, the impact of inter-lineage LASV diversity on previously mapped B-cell and T-cell LASV GP epitopes in the literature is explored in this review, and the consequences of LASV diversity for LASV vaccine development is discussed as well.

## 2. LASV Lineages, Diversity, and GP Epitopes 

### 2.1. GP Variation among LASV Lineages

The GP trimer, the main surface antigen of LASV, is a target of host antibodies and T-cells [48,53]. This GP “spike” has been suggested to be under host immune selective pressure leading to high intra-host LASV diversity [12]. The GP is also regarded as the primary immunogen for LASV vaccine development [36] because vaccines expressing LASV GP conferred a higher level of protection than those expressing LASV NP only [23,47]. GP1 is the most variable subunit, while the GP2 is the most conserved subunit among mammarenaviruses [14,54,55]. Synonymous single nucleotide variations (SNV) were found in similar rates in both subunits of the GP and most non-synonymous SNVs were found in the GP1 subunit among intra-host LASV strains [12]. Thus the GP1 subunit appears to be more tolerant to mutations, whereas the GP2 subunit is less tolerant to variations probably because it contains important loci for entry, fusion, and transmembrane anchoring [48].

Among LASV confirmed and proposed lineages (I-IV, Mali, Togo, and Kako strains) [11,14,16,56,57,58], the inter-lineage variation in the amino acid (AA) sequence of the GP is between 4.9–11.0%. This is higher than previously reported values based on five lineages [42], a likely result of the increasing diversity of LASV. Table 1 shows the GP amino acid identity matrix of confirmed and proposed LASV lineages. KAK-428 GP is the most divergent. This LASV diversity may affect GP epitopes recognized by the immune system. Anderson et al. showed that most non-synonymous iSNVs occurred within predicted GP B-cell epitopes, and GP T-cell epitopes also harbor some non-synonymous iSNVs but they appear to be more conserved than B-cell epitopes [12].

### 2.2. GP Epitope Variation among LASV Lineages

In a study by Robinson et al., LASV GP B-cell epitopes were mapped using 113 human monoclonal antibodies (mAbs) derived in vitro from the blood of LF survivors, 15 from Sierra Leone and 2 from Nigeria [53]. This study generated LASV GP-specific mAbs, characterized their neutralization properties, classified them using cross-competition assays, and determined the cross-reactivity of the mAbs among LASV lineages and other arenaviruses. Their putative epitopes were mapped and lastly, the degree of germline divergence of the mAbs was assessed. MAb neutralization property was assessed using assays of HIV pseudo-virus expressing LASV GP of Lineage I–IV, LCMV pseudo-virus expressing LASV GP Lineage IV, and plaque reduction neutralization test (PRNT) using the real Lineage IV LASV Josiah strain [53].

The results showed various neutralization patterns across the pseudo-virus neutralization assays with largely similar patterns between the two LASV Lineage IV pseudo-viruses (HIVpp and LCMVpp) [53]. However, the PRNT required a higher 50% inhibitory concentration (IC_50_) and 80% inhibitory concentration (IC_80_) compared to both pseudo-virus assays, with more mAbs becoming weakly potent or losing activity in the PRNT assay [53]. Since the PRNT was performed against Lineage IV only, the PRNT activity of the mAbs against authentic LASV Lineage I–III virus may be different from the pattern seen in the pseudo-virus assays, therefore, the true LASV inter-lineage variation of mAb neutralization was not directly demonstrated. This could mean that the inferred therapeutic potency of some mAbs against LASV Lineage I–III may be in doubt. Furthermore, Cross et al. showed that in vitro mAbs potency against LASV did not correlate with protection in vivo in the mAb therapeutic study of guinea pigs, where some highly potent mAbs in vitro provide less protection in vivo [52]. As the viral challenge was only the LASV Lineage IV Josiah strain in the mAb therapy study, the correlation of broad cross-reactivity of MAbs in vitro with possible “pan”-lineage therapeutic efficacy across LASV lineages is unknown. The limitations of cost, availability of live LASV strains, and logistics of safety using BSL4 facilities for multi-lineage authentic LASV PRNT and challenge experiments may be a reason for using only one lineage in these studies. Moreover, cross-reactivity assays (using HEK293T cells transfected with eukaryotic expression vectors encoding GPs of different LASV lineages) predominantly demonstrated a similar pattern of mAb binding seen in the pseudo-virus assays in this study by Robinson et al. [53], demonstrating a relatively consistent pattern of mAb binding across LASV lineages.

The majority of the mAbs targeted conformational epitopes on GP1, GP2, or both subunits on the GP, and only seven mAbs targeted linear epitopes on the GP2 subunits [53]. A total of 16 were neutralizing mAbs, which exhibited higher binding affinities and huge germline divergence, and 13 of these in the GPC-B cross-competing group were directed against conformational epitopes on both GP1 and GP2 in the glycoprotein complex configuration [53]. The main conformational epitope, bound by 37.7H mAb, was later structurally defined to occupy the surface of two GP monomers close to the base of the GP trimer. Here it binds four separate regions, two regions each in site A and B (Figure 3) [32]. Site A comprises 62–63AA residues of the GP1 N-terminal loop and 387–408AA sequence in the T-loop and HR2 of GP2, while Site B consists of 269–275AA sequence of the fusion peptide and 324–325AA sequence of the HR1 of GP2 [32]. The recognition of this conformational epitope is fairly conserved between 37.7H and two other GPC-B MAbs, 18.5C and 25.6A where they share a similar footprint on the GP trimer in both sites A and B [38].

The GP1 62–63AA sequence of site A is fairly conserved among LASV lineages, but it is surrounded by poorly conserved amino acid residues within 59–65AA sequence. Site B 269–275AA sequence contains the highly variable 272–274AA sequence, and its 324AA residue is also radically substituted among these LASV lineages (Figure 4). Even though there are inter-lineage variations within and around sites A and B, it is possible that their impact on the conformation of this epitope and antibody binding may not be significant because there are multiple regions of the GP1 and GP2 comprising this epitope. Furthermore, Hastie et al. recently showed that well-conserved residues Q405 and D408 in GP2 site A are most important in GPC-B mAb binding and complete dual-site epitope binding is not required for mature GPC-B mAb binding [38]. However, authentic LASV Lineage I virus and its rVSV-LASVGP pseudo-virions were refractory to neutralization by GPC-B mAbs 37.7H and 25.6A [38], even though 37.7H mAb is broadly cross-reactive against the GP of LASV lineages I–V as well as LCMV on eukaryotic expression vectors and HIV-based pseudovirions [53]. This poor neutralization of LASV Lineage I may reflect the impact of LASV diversity on this epitope, given that the 37.7H and 25.6A mAbs were derived from Sierra Leonean survivors who were exposed to LASV Lineage IV. Based on the cross-reactivity assays, the 37.7H epitope most likely has a well-conserved quaternary conformation, fortunately, protection does not seem to depend on its neutralizing potency. The prefusion configuration of LASV GP trimer is required for 37.7H mAb binding and it stabilizes this conformation to prevent GP fusion, viral entry, and infection [32]. The GPC-B mAbs also interact with well-conserved glycans at N390, N395, and N79, albeit this interaction serves to diminish antibody neutralization potency [38].

On the other hand, another cross-competing group of mAbs that bind in the GP trimeric conformation demonstrated variable neutralization results against LASV lineages, and they did not cross-react against other mammarenaviruses. These GPC-A mAbs putatively target the regions 62–68AA on GP1 and 270–278AA on GP2 in a conformational epitope [53]. 36.1F of this group, neutralized LASV Lineage IV pseudo-virion and intact virion, but it was inactive against LASV Lineage I–III pseudo-virions [53]. Both GP1 and GP2 regions of the putative epitope contain poorly conserved residues among LASV lineages, especially 272–274AA sequence which is relatively conserved among Lineage I–III but divergent in Lineage IV and others (Figure 4). Therefore, 36.1F mAb, which was derived from convalescent plasma of Sierra Leone origin [53], may be very specific against the 272–274AA residue of Lineage IV only, suggesting that the 36.1F/GPC-A epitope is affected by the LASV lineages’ diversity.

The remaining three neutralizing antibodies, members of GP1-A MAb group, were directed against a conformational epitope on residues 111–117 of the GP1 subunit only [53]. Two of these mAbs were cross-reactive against LASV Lineage I-IV and LCMV while the last, 19.7E MAb, did not neutralize LASV Lineage III and LCMV. Subsequent modification of the 112–114AA sequence in LASV Lineage IV from IIN to LLN of Lineage III markedly reduced the neutralization activity of 19.7E and 10.4B GP1-A mAbs [53]. This confirms that the poorly conserved residues 112–114AA of the GP1 among LASV lineages would impact the GP1-A conformational epitope. Andersen et al. demonstrated a similar outcome in which LASV intra-host variants with minor alleles at position 89AA and 114AA significantly diminished binding by GP1-A mAbs [12]. These results are evidence of LASV diversity affecting GP epitopes.

All mAbs that bound only the LASV GP2 subunit from this study were non-neutralizing mAbs, and they were mostly directed against the conformational epitope located in the well-conserved 328–358AA GP2 sequence, and some of these had limited cross-reactivity against other arenaviruses and one mAb (6.6C GP2-B) neutralized LASV Lineage I and IV only [53]. All the GP2-linear-epitope-binding mAbs, which target highly conserved sequences (GP2-L1 mAbs binding 300–315AA residues, GP2-L2 mAbs binding 361–375AA residues of T-loop, and GP2-L3 mAbs binding 401–415AA HR2 motif of GP2 subunit), were cross-reactive against LASV lineages I through IV. GP2-L1 and GP2-L2 mAbs also widely cross-react with other mammarenaviruses [53]. Given that the GP2 subunit is the least variable portion of the GP sequence, it is not surprising that these epitopes are largely unaltered by LASV lineage diversity. Another group of non-neutralizing mAbs, GP1-B mAbs, binds a conformational epitope mapped to the well-conserved 119–134AA region of the GP1 subunit only [53]. In summary, most of the non-neutralizing mAbs targeted conserved epitopes on the GP1 and GP2 subunits individually.

In a recent study by Amanat et al., cross-reactive GP2-binding mAbs against Old World and New World arenaviruses were non-neutralizing in vitro with minimal antibody-dependent cellular cytotoxicity (ADCC) activity and were non-protective against LASV disease in mice [54]. These GP2-binding mAbs did not neutralize LASV because they probably targeted the post-fusion GP2 conformation, therefore, they were unable to inhibit viral entry. In vivo protection was not extensively tested in this study in which few cross-reactive GP2 mAbs were raised, and the lack of protection conflicts with a prior report by Ruo et al. demonstrating the neutralizing property of GP2 specific mAbs that bind conserved epitopes on GP2 subunit of Old World and New World arenaviruses. Furthermore, recent vaccine studies have also shown that non-NAbs may contribute to protection against clinical disease in guinea pig and NHP models [50,51,62]. However, the GP binding site of the vaccine-induced non-NAbs were not mapped in these studies. Taken together, one may speculate that non-NAbs which are produced early in LASV infection may be targeting more conserved epitopes in the GP1, GP2 subunits, just like the GP1-B, GP2-B, and GP2-L1–L3 mAb groups, and they may prove to be a useful direction for vaccine research. In addition, the GP epitopes important for protection against clinical disease (non-NAbs) may differ from those necessary for sterilizing immunity (NAbs) and this may also influence the type of protection and the choice of epitope for vaccine development. Induction of non-NAbs would be an important consideration of future LASV post-exposure prophylactic or therapeutic vaccines.

The importance of T-cell mediated protection from LF underlies the focus of some research on LASV T-cell epitopes and T-cell inducing LASV vaccines. In a study by Botten et al., human leucocyte antigen HLA-A2 restricted CD8+ T-cell epitopes of the GP protein were mapped and they predicted putative epitopes that bind the HLA-A2 supertype family (which is present in 50% of human population regardless of ethnicity) using the HLA motif algorithm, and subsequently screened peptides of these epitopes downstream for immunogenicity in HLA transgenic mice, functional avidity of CD8+ T-cell responses, and HLA restricted human antigen presenting cell (APC) processing of these peptides [49]. Three HLA-A2 restricted CD8+ T-cell epitopes of the GP protein were identified, and their linear epitopes (42–50AA–GLVGLVTFL, 60–68AA–SLYKGVYEL, and 441–449AA, YLISIFLHL) were isolated from the Lineage IV Josiah Strain, and these peptides were used to vaccinate HLA-A*0201 transgenic mice, followed by challenge with recombinant vaccinia virus expressing the Josiah Strain LASV GP (rVV/LASV-GP) [49].

Even though all the nonapeptides induced robust CD8+ T-cell response, the 441–449AA peptide did not prevent rVV/LASV-GP replication whereas the other two nonapeptides inhibited rVV/LASV-GP replication of the same LASV strain. Cross-reactivity tests of CD8+ T-cells specific for the Lineage IV Josiah strain peptides against same sequence nonapeptides of Lineage III GA391 strain showed that the GA391 variant 59–67AA peptide was not recognized by CD8+ T-cells specific to the equivalent Josiah 60–68AA peptide (Table 2) [49]. The GP1 60–68AA peptide sequence has a radical replacement of 60AA residue, and it is variable in the 61AA and 65AA residues between the Josiah and GA391 strains (Table 2) [49]. This region is also poorly conserved among other LASV lineages, a consequence of LASV diversity of this T-cell epitope (Figure 4) and CD8+ T-cell epitope specific immunity by extension. Conversely, the GP2 441–449AA sequence is highly conserved among LASV lineages, while the other GP1 42–50AA sequence harbors minor variations.

Other studies have predicted more conserved T-cell epitopes (Table 3) [48,63,64,65]. The GP2 289–301AA CD4+ T-Cell epitope is highly conserved among LASV lineages, and all other Old and New world mammarenaviruses. The epitope overlaps with the GP fusion peptide, which may constrain mutations at this site [48]. Among the computer-predicted T-cell epitopes, the GP1 210–218AA epitope appears to be most promising. However, these computer-predicted T-cell epitopes need to be tested in vitro to determine the APCs processing and T-cell binding properties. Sullivan et al. recently showed a cross-reactivity of LASV-specific T-cells of Nigerian LF survivors (Lineage II and III LASV infections) to GPc epitopes in peptides derived from Lineage IV LASV (Josiah strain) [66]. Cross-reactive CD8+ T-cell responses were mostly directed to well-conserved GPc epitopes in peptides spanning 240–259AA and 412–451AA, especially the 440–449AA epitope whereas CD8+ T-cell responses to peptides from less conserved amino terminal of GP1 (1–40AA, 34–58AA, and 58–82AA) were seen only in Sierra Leonean LF survivors (Lineage IV LASV infections) [66].

## 3. Implications of LASV Diversity on Vaccine Development

LASV lineage diversity is reflected in a variety of B and T-cell GP epitopes. Quaternary epitopes that combine multiple sites in different subunits of the GP complex are broadly cross-reactive across LASV lineages [32,53] however, the cross-reactivity was measured in LASV/HIV-1 pseudo-virus neutralization assays which are artificially sensitive to neutralization and poor mimic of authentic LASV PRNT. GP2 epitopes also appear well conserved. While, poorly conserved epitopes are commonly found on the GP1 [49,53]. These poorly conserved epitopes decrease the breadth of antibody binding across LASV lineages, a task that is already constrained by glycan shielding of the GP trimer therefore, they pose a challenge for pan-Lassa vaccine development.

The World Health Organization (WHO) recognizes the challenge presented by LASV lineage diversity [6]. In its Target Product Profile (TPP) for a LASV vaccine, it stresses that preferred LASV vaccine candidates must protect against LASV Lineage I–IV, with a high priority for development being preventive use, in addition to other criteria [68]. The increase in LASV diversity, with the emergence of three additional proposed lineages, continually complicates LASV vaccine development. It is important to continue monitoring the mutational spectrum of LASV in order to detect the emergence of new lineages and incorporate changes to vaccines to cover the extent of LASV diversity [12]. It is also beneficial to improve the surveillance for LASV strains across West Africa. Since LASV spread centuries ago, novel strains and lineages may be lurking in parts of the region just as new but evolutionarily ancient lineages have been discovered within the past decade.

Epitope-based vaccines are an attractive method for LASV vaccine development as they induce HLA-restricted protection with the flexibility of presenting either subdominant or immunodominant epitopes with equal effectiveness, while avoiding immunosuppressive epitopes [49]. They can be affected by LASV diversity when poorly conserved epitopes, such as GP1 60–68AA, are utilized. This can be circumvented in the early stages by extensively testing the epitope-specific T-cell cross-reactivity across all LASV lineages. Whereas, well-conserved GP2 T-cell epitopes across Old and New World arenaviruses may hold the promise of an universal mammarenavirus vaccine [48]. Even so, epitope-based vaccines still face the HLA restriction hurdle. Most of the HLA alleles predicted to present these epitopes are found in variable proportions in the endemic region, ranging from 12.69% to 68.7% in West Africa [48,63,64,65].

Another approach is to present whole LASV GP complex on vaccine constructs (e.g., viral vectors, virus-like particles (VLP), and replicons), as practiced in most of the successful LASV vaccine candidates [23,46,50,51,69]. This presents both variable and conserved GP subunit epitopes, as well as quaternary GP epitopes, to induce B-cell and T-cell mediated protection. It is important to present the GP ectodomain in its prefusion conformation in order to present quaternary epitopes in the appropriate conformation for antibody binding. This may increase the chance of eliciting protective antibodies which acts by inhibiting receptor attachment, viral membrane fusion, or both, because conformation epitope binding mAbs (GP1 and GPC) with similar actions have been shown to be very effective [32,45,52,53]. In contrast, the presentation of the post-fusion GP conformation would likely elicit unprotective antibodies, as inferred from the study by Amanat et al. [54]. GP prefusion ectodomain conformation may also be necessary to achieve protection without symptoms, as seen in the LASSARAB study [50] however, it is unclear which aspect of this vaccine’s design, potent Glucopyranosyl Lipid A in stable emulsion GLA-SE addition, vaccine regimen, or animal immune response is responsible for asymptomatic protection observed among guinea pigs. Furthermore, LASV GP prefusion ectodomain can be engineered to induce variants of GPC-B antibodies that possess a tri-arginine patch in the 37.7H site A epitope-binding-CDR of the heavy chain, mAb 18.5C-like kappa light chains and require few somatic mutations, as shown by Hastie et al., to have increased breadth of neutralization across LASV lineages I–V. In addition, site-selective deglycosylation of LASV GP at N390 and N395 may also improve its immunogenicity [38]. This may generate protective NAbs earlier post-vaccination with a potential for pan-Lassa sterilizing immunity.

The expression of both LASV GP and NP has been shown to be required for efficient protection with near sterilizing immunity in NHPs [21,47,51,70]. Additional LASV NP expression in the ML29 vaccine is suggested to be critical for extending cross-protection from LCMV which has a 50% NP sequence homology with LASV [27]. It likely contributes additional conserved epitopes to induce broadly protective cellular immune responses. In human LF survivors, CD4+ T cell responses were mainly directed towards LASV NP, and anti-LASV NP CD4+ T-cell responses were broadly cross-reactive, while cross-reactive anti-LASV NP CD8+ T-cell responses target relatively conserved NP epitopes [66,71]. However, vaccines expressing only LASV NP have not shown effective protection in NHPs [47], despite a high potency in guinea pigs vaccinated with recombinant modified vaccinia Ankara virus (MVA), recombinant vaccinia virus, or alphavirus replicons expressing LASV NP [23,72,73,74]. Even so, LASV infection in guinea pig models weakly correlates with human clinical disease outcomes [27]. Even though, vaccine-induced immunity appears dependent on the expression vector, the co-expression of LASV GP and NP may also contribute to induction of sterilizing immunity, the “Holy Grail” of vaccine-conferred protection. NP induces effective viral control at the early stage of arenavirus infection since in the absence of NP synthesis in vivo, there are no detectable CD8+ T-cell responses and protective immunity [75,76]. Sterilizing immunity will be essential to prevent LASV transmission from vaccinated individuals. It is also speculated to prevent immune-mediated neurologic sequelae of LASV infection in animal models, such as sensorineural deafness, that is due to persistent viremia in immunologically privileged tissues [51,77,78,79] however, there is no evidence of immune-mediated pathology in human LF so far.

Development of region-specific LASV vaccines has also been suggested as an approach to surmount LASV diversity, which would be based on the molecular epidemiology of LASV lineages [12]. Since it was found that no specific LASV lineage was driving the 2018 Lassa outbreak in Nigeria and LASV lineages appear relatively stable in their geographic clustering [80], developing region-specific LASV vaccines is plausible. However, LASV lineages I-III are endemic in Nigeria, meaning a “tri-valent” vaccine or “uni”-valent vaccine protecting against three lineages would be needed in Nigeria. From another perspective, this means three-quarters of all the confirmed lineages should be covered by one Nigeria-specific vaccine. Furthermore, Lineage II and III are phylogenetically distant from one another and Lineage II has roughly a similar phylogenetic distance from Lineage III in Nigeria and Lineage IV in the MRU region [12,56,80]. Therefore, the method of developing a single Nigeria-centric LASV vaccine protective against three lineages would likely be no less different than that for generating a “pan”-Lassa vaccine. This region-specific approach also discounts the contribution of human-to-human transmission (though minor) which can spread LASV lineages beyond the endemic region, via human transportation. Ultimately, the presence of some well-conserved subunit and conformational prefusion LASV GP epitopes, as well as the outcomes of some advanced LASV vaccine studies, makes a single “pan”-Lassa vaccine potentially attainable without developing region-specific variants. Therefore, the region-specific approach is less favored, as evident in the WHO criteria.

Many LASV vaccine candidates have undergone preclinical “proof of concept” efficacy trials in animals with variable results (Table 4) [27]. Most of these candidates contain the LASV GP antigen from the Lineage IV-Josiah strain and only homologous challenge was performed in vaccinated animals [27,42], leaving little evidence of the breadth of their immune protection across LASV lineages. Poorly understood and inconsistent virulence of LASV lineages in animal models is also an obstacle for providing evidence of an universal LASV vaccine [81]. For example, the mild nature of LASV Soromba-R (proposed lineage V) strain’s disease in NHPs weakens the evidence of protection conferred by vaccines in the cohort of NHPs challenged with LASV Soromba-R strain [23,25]. Two advanced vaccine candidates were suggested for accelerated research and development by global health leaders in vaccinology: ML29 and rVSVΔG/LASV-GP [82]. In addition, a Measles virus-vectored LASV vaccine (MeV-NP or MV-LASV) and a DNA-based LASV vaccine (INO-4500) have advanced to phase I clinical trials [83,84,85].

The ML29 vaccine candidate is a reassortant virus of the L RNA related from the Mopiea virus (MOPV) and S RNA from Lineage IV/Josiah Strain LASV, which prevented mortality in marmosets and guinea pigs challenged with the LASV Lineage IV/Josiah strain and LASV Lineage II/803213 strain respectively with a potential for sterilizing immunity [21,69,70]. cDNA clones have been used recently to produce a recombinant ML29 (rML29) to solve the challenge of LASV diversity. This would extend the breadth of cross-protection by expressing another LASV lineage’s antigen utilizing an “arenavirus tri-segmented (r3)” platform to produce r3ML29 (with S RNA from Lineage I and IV) [27]. The rVSV-vectored vaccine candidate, expressing LASV Josiah strain’s GP (rVSVΔG/LASV-GP), protected guinea pigs and macaques’ cohorts from fatal disease post-challenge with two different LASV Lineage IV strains without viremia, furthermore guinea pigs were also protected against Soromba-R strain (proposed lineage V strain) and Lineage I LP strains respectively [23]. Another attenuated rVSV-vectored LASV vaccine expressing the Josiah strain’s GP (rVSV-N4ΔG-LASVGPC), in a quadrivalent Vesiculovax vaccine formulation comprising three other rVSV(N4CT1)-based filovirus vaccines, also protected against lethal LASV Lineage II heterologous challenge in NHPs [86]. These suggest that broad cross-protection across LASV lineages can be achieved using a vaccine expressing LASV GP of one lineage. However, protection against heterologous LASV challenge demonstrated in guinea pigs should be taken with caution, given that this LASV vaccine-induced protection poorly correlates with NHPs and possibly humans. For example, VEEV-TC83 RNA replicon particles had reduced efficacy from guinea pigs to NHPs in homologous LASV Lineage IV vaccine/challenge studies, and poor protection against a heterologous LASV lineage challenge in NHP [27,74,87].

More recently, the Measles virus-vectored LASV vaccine (MV-LASV) expressing both LASV GP and NP, fully protected macaques against the lethal homologous LASV Lineage IV Josiah strain challenge without viremia [51]. The LASV GP-expressing attenuated recombinant MOPV-based vaccine (MOPEVAC_LASV_) also showed comparative full protection in this study, however, there was transient low-titer viremia following the same LASV challenge [51]. The MV-LASV has progressed to phase I clinical trials, although its breadth of protection remains to be demonstrated in NHPs. It is thought that MV-LASV’s “effectiveness” may be hampered by pre-existing immunity in humans, however, the immunogenicity of a Chikungunya virus vaccine on the same platform was not affected by pre-existing anti-measles antibodies in humans [27,89]. Another DNA-based LASV GP-expressing vaccine (pLASV-GP or INO-4500) is also in phase I clinical trials. It gave full protection without clinical signs of illness or viremia in NHPs challenged with homologous LASV Lineage IV Josiah strain [78,79]. The breadth of protection for these vaccines remains to be evaluated.

Adjuvants, such as GLA-SE and related-TLR-4 adjuvants, may represent a unique potential for LASV vaccine development. GLA-SE improved the immunogenicity of the inactivated LASSARAB vaccine in guinea pigs [50] and has been shown to improve the breadth and functionality of both humoral and cellular immune response with other antigens such as influenza and HIV [90,91]. There is hope that it may contribute to overcoming LASV diversity. These adjuvants may pave the way for another approach to LASV vaccine design using inactivated, sub-unit, and epitope-based vaccine constructs. These adjuvanted non-replicating vaccines would be important to extend the protection of LASV vaccines to the most vulnerable group such as pregnant women, who have a high-mortality rate from Lassa fever but are not recommended by the FDA to receive replication-competent vaccines. Furthermore, adjuvants could also make non-replicating vaccines more attractive in West Africa because of their potential safety in HIV-infected individuals.

Since there is no licensed LASV vaccine and other measures of prevention and control of LF are needed in the endemic regions. This includes prevention of rodent-to-human transmission, prevention of human-to-human transmission in the community, and healthcare settings as well as early laboratory diagnosis and treatment.

## 4. Conclusions

LASV diversity is evident in the primary sequence of its GP, and it is reflected in the LASV GP’s B-cell and T-cell epitopes to varying degrees. This needs to be considered in the LASV vaccine design. Differential protection against LASV lineages, seen in some vaccine/challenge animal studies such as the VEEV-TC83 RNA replicon particles studies [27] and the poorly conserved GP1 60–68 AA peptide (Botten et al. [49]), makes the dearth of evidence on broad protection in NHPs against heterologous LASV lineage challenges concerning.

Engineering the expressed LASV GP to efficiently induce variant broadly-neutralizing GPC-B group of antibodies and specific T cell response, while maintaining the pre-fusion configuration to present the appropriate quaternary-structure epitopes, may increase the breadth of protection. Given the importance of cell-mediated immune protection in LF that is relatively biased towards LASV NP and evidence of cross-reactive T-cell responses in LF survivors, it seems that vaccines co-expressing LASV GP and NP of a single lineage (Lineage IV/Josiah strain) will more likely induce broader cross-protection against other LASV lineages with a potential for sterilizing immunity.

Furthermore, LASV vaccines need to be tested for protection against all LASV lineages in the gold standard NHP model. With the atypical pathogenicity of the proposed lineage V strains in NHPs, this Malian lineage distinguishes itself despite its close relationship to Lineage IV, hence the virulence of other newly proposed lineages should be tested in animal models. The proposed lineages should also be considered in vaccine/challenge studies and the WHO may need to review its Target Product Profile for LASV vaccines to reflect the need for cross-protection beyond the established four LASV lineages.

## Figures and Tables

**Figure 1 viruses-12-00386-f001:**
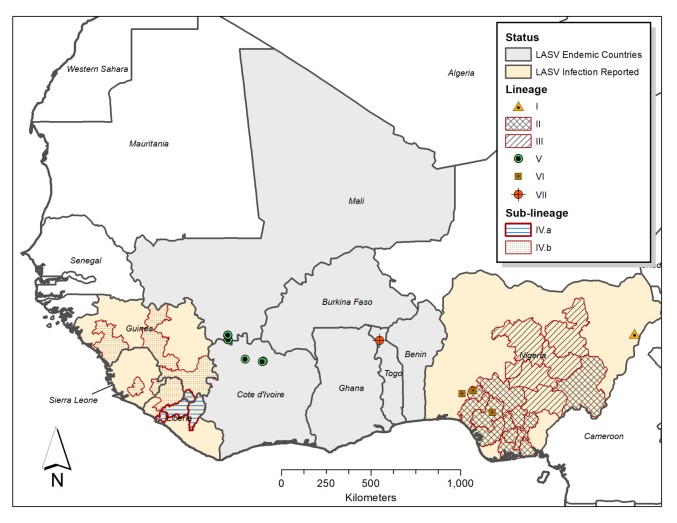
Geographical distribution of Lassa virus (LASV) Lineages in West Africa. The shaded areas and symbols on the nap show the distribution of LASV lineage and sub-lineage infections as observed in West Africa, based on data from Ehichioya et al., 2019 and Wiley et al., 2019 [17,18]. The shaded areas depict administrative regions/states within LASV endemic countries, where each represented LASV lineage (II and III) or sub-lineage (IV.A and IV.B) has been reported. The symbols show the locations where LASV lineages I, V, VI, and VII have been observed. Note: Shaded areas and symbols do not represent the incidence or prevalence of LASV infections in each location or administrative region.

**Figure 2 viruses-12-00386-f002:**
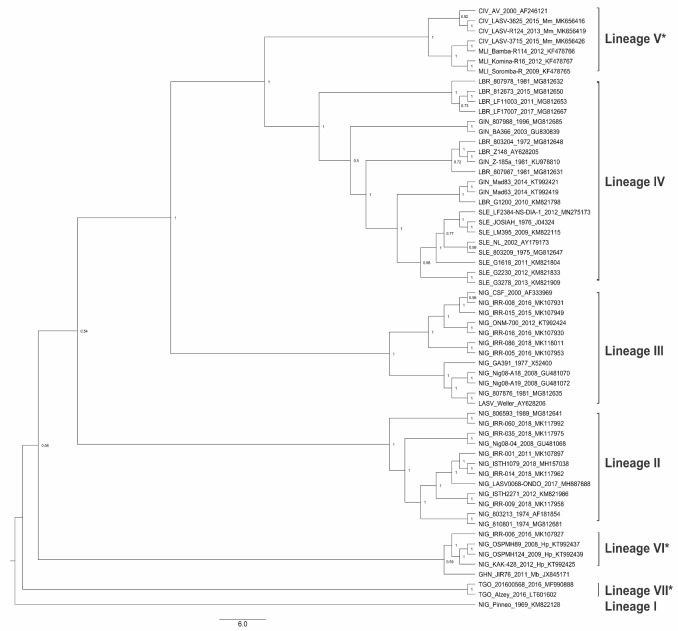
Maximum clade credibility tree of LASV Lineages’ glycoprotein (GP) gene. A phylogenetic analysis of the GP gene of members of confirmed and proposed (*) LASV lineages. The tree is rooted in Lineage I and each node shows the Bayesian posterior probabilities support. The tree tips are labelled to shown country, strain, year of collection, and GenBank accession numbers. Precursor glycoprotein (GPC) nucleotide sequences with greater than 60% coverage for the LASV GPC coding sequence (CDS) were retrieved from the GenBank, a maximum likelihood tree was generated in RAxML v8.2.12 [39] on CIPRES [40]. Bayesian phylogenetic inference was performed using BEAST v1.10.4 [41] with Generalized Time Reversible (GTR) plus gamma substitution model, uncorrelated relaxed clock model in lognormal distribution, SkyGrid coalescent tree prior setting, and the maximum likelihood tree as starting tree. Markov chain Monte Carlo (MCMC) chains were run for 250 million iterations, sampled every 10,000 states and 2500 trees were discarded as burn-in, to obtain an effective sample size (ESS) >200 for all parameters. Maximum clade credibility tree was drawn in Tree Annotator v1.10.4 [41].

**Figure 3 viruses-12-00386-f003:**
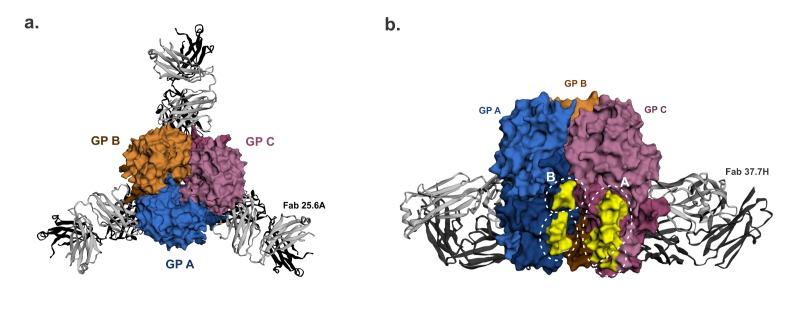
GPC-B† quaternary structure epitope. The 25.6A and 37.7H monoclonal antibodies (mAbs) were raised to the LASV GPC-B conformational epitope. (**a**). (Top view) The LASV GP trimer is bound by three 25.6A Fabs, with each Fab binding two GP monomers near the trimer’s base. (**b**). (Front view rotated 60°) The GP trimer bound near its base by three 37.7H Fabs (two Fab in view) in a similar manner as above. Amino acid residues in the 37.7H epitope sites A and B highlighted in yellow to show the antibody footprint. GP monomers are shown as surface representations. LASV GP-mAb bound structures were retrieved from the Protein Database using PDB IDs 6P95 and 5VK2 for 25.6A mAb-bound GP and 37.7H mAb-bound GP respectively, image generated in EzMol 2.1 online program [60] and labelled manually.

**Figure 4 viruses-12-00386-f004:**
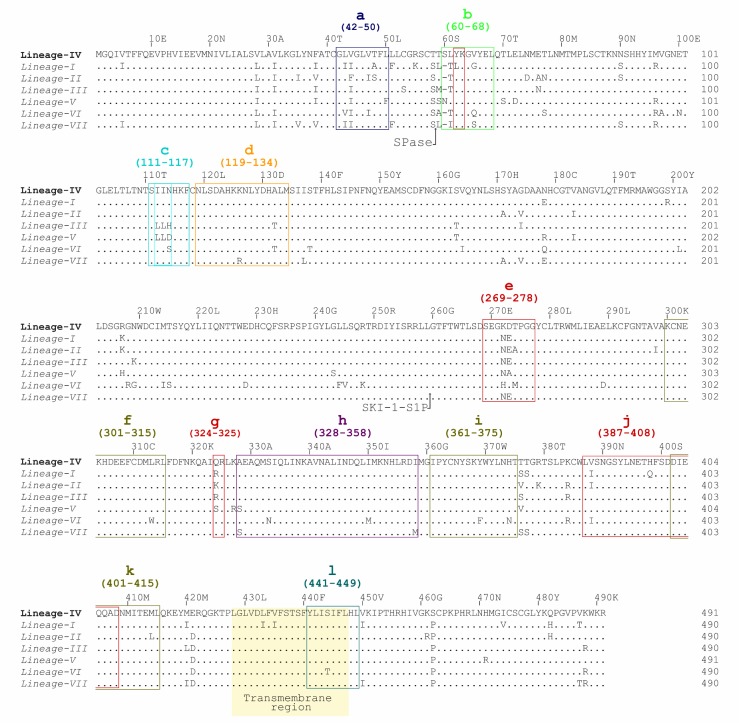
Aligned GP precursor amino acid sequence of LASV confirmed and proposed lineages. Multiple sequence alignment of prototype LASV strains selected from the literature (see Table 1 for detailed description) created with Clustal Omega in Jalview 2.11.0 [61] and manually annotated. LASV lineages and strains: Lineage I—Pinneo 1969 Strain, Lineage II—Nig08-04 Strain, Lineage III—Nig08-A19 Strain, Lineage IV—Josiah Strain, proposed lineage V—Soromba-R strain, and proposed lineage VII—Alzey strain), and proposed lineage VI—KAK-428 strain. LASV GP B-cell epitopes include: 37.7H epitope site A (*b & j*) and site B (*e & g*), GPC-A (*b & e*), GP1-A (*c*), GP2-B (*h*), GP2-L1(*f*), GP2-L2 (*i*), and GP2-L3 (*k*). LASV GP T-cell epitopes include: GP1 epitope 42–50AA (*a*), GP1 epitope 60–68AA (*b*), and GP2 epitope 441–449AA (*l*).

**Table 1 viruses-12-00386-t001:** GP amino acid (AA) percent identity matrix among confirmed and proposed LASV lineages.

	Lineage I Pinneo 1969	Lineage II Nig08-04	Lineage III Nig08-A19	Lineage IV Josiah	Lineage V * Soromba-R	Lineage VI * KAK-428	Lineage VII * Alzey
**Lineage I** Pinneo 1969	100.00	92.24	92.65	93.06	91.63	88.98	95.10
**Lineage II** Nig08-04	92.24	100.00	93.27	93.67	92.04	89.80	92.65
**Lineage III** Nig08-A19	92.65	93.27	100.00	94.90	94.69	91.22	93.47
**Lineage IV** Josiah	93.06	93.67	94.90	100.00	94.50	91.63	93.67
**Lineage V** * Soromba-R	91.63	92.04	94.69	94.50	100.00	89.39	92.45
**Lineage VI** * KAK-428	88.98	89.80	91.22	91.63	89.39	100.00	89.80
**Lineage VII** * Alzey	95.10	92.65	93.47	93.67	92.45	89.80	100.0

(*) Proposed Lineages—created by Clustal 2.1. A PubMed database search (https://www.ncbi.nlm.nih.gov/pubmed) was performed and prototype strains of each LASV lineage were selected from key literature which had previously utilized these strains to assess LASV diversity, inter-lineage antibody cross-protection, or originally published the proposed lineages. Primary sequence of the glycoprotein of each LASV prototype strain was downloaded from the protein database (https://www.ncbi.nlm.nih.gov/protein); they were aligned using Clustal Omega hosted online on the European Bioinformatics Institute (EBI) website (https://www.ebi.ac.uk/Tools/msa/clustalo/) (see Figure 4), and the amino acid sequence identity matrix was generated. LASV Lineage/Prototype Strain sources: Lineage I—Pinneo 1969 Strain: Gen-Bank Ascension Number: KM822128.1—Ref [12]; Lineage II—Nig08-04 Strain: Gen-Bank Ascension Number: GU481068.1—Ref [56]; Lineage III—Nig08-A19 Strain: Gen-Bank Ascension Number: GU481072.1—Ref [56]; Lineage IV—Josiah Strain: Gen-Bank Ascension Number: NC_004296.1—Ref [57]; Soromba-R Strain (proposed lineage V): Gen-Bank Ascension Number: AHC95553.1—Ref [59]; Alzey Strain (proposed lineage VII from Togo): Gen-Bank Ascension Number: LT601602.1—Ref [16]; KAK-428 Strain (proposed lineage VI): Gen-Bank Ascension Number: KT992425.1—Ref [11].

**Table 2 viruses-12-00386-t002:** LASV CD8+ T-cell epitopes and inter-lineage cross-reactivity.

Peptide *^a^*	LASV Strain (Lineage)	Peptide Sequence *^b^*	Functional Avidity (M) *^c^*
GP1 42–50AA	Josiah (IV)	GLVGLVTFL	6 × 10^−11^
GP1 42–50AA	GA391 (III)	* * **I** * * * * * *	6 × 10^−11^
GP1 60–68AA	Josiah (IV)	SLYKGVYEL	6 × 10^−11^
GP1 59–67AA	GA391 (III)	**LI** * * * **T** * * *	*Not immunogenic*
GP2 441–449AA	Josiah (IV)	YLISIFLHL	6 × 10^−11^
GP2 441–449AA	GA391 (III)	* * * * * * * * *	6 × 10^−11^

(^a^) Peptide amino acid residues within LASV GP. (^b^) Letters in boldface denote amino acid substitutions between the LASV Lineage III strain and LASV Lineage IV strain. (^c^) Cross-reactivity of CD8+ T cells isolated from HLA-A*0201 transgenic mice immunized with LASV Lineage IV strain Josiah peptide at 11 to 14 days post-immunization and their interferon-gamma (IFN-γ) response to peptide-pulsed JA2.1 target cells that received decreasing gradient doses of either LASV Lineage IV—strain Josiah or Lineage III—GA391 peptide. Each value represents the endpoint quantity of peptide required to generate a statistically significant IFN- γ response. Modified from: Botten, J.; Alexander, J.; Pasquetto, V.; Sidney, J.; Barrowman, P.; Ting, J.; Peters, B.; Southwood, S.; Stewart, B.; Rodriguez-Carreno, M.P.; et al. Identification of protective Lassa virus epitopes that are restricted by HLA-A2. Journal of virology 2006, 80, 8351–8361.

**Table 3 viruses-12-00386-t003:** Other LASV-specific T-cell epitopes.

GP Subunit	T-Cell Epitope	Consensus Peptide Sequence	Comments	Reference
GP2	289–301AA	ELKCFGNTAVAKC	Highly conserved CD4+ T-cell epitope on the GP2 N-terminal fusion domain. Bound to limited number of HLA whose alleles occur <25% of Africans and Caucasians.	Meulen et al. [48]
GP1	210–218AA	WDCIMTSYQ	Computer predicted nonapeptide epitope; highly conserved among LASV lineages. Predicted to bind 64 major histocompatibility complex MHC molecules; however, global distribution of these MHCs is not stated.	Verma et al. [63]
GP1	41–49AA	SSNLYKGVY	Computer predicted nonapeptide epitope. Mismatched peptide number and sequence: 41–49AA corresponds to CGLIGLVTF, which is well conserved; while SSNLYKGVY is found within the poorly conserved 58–66AA residues Predicted to interact with 17 HLA-I and 16 HLA-II proteins which only cover 68.17% of the endemic region—West Africa.	Hossain et al. [64]
GP1 C-terminal and GP2 N-terminal	258–266AA	LLGTFTWTL	Computer predicted epitope; highly conserved among LASV lineages. It lies within the SKI-1-S1P cleavage site (RRLL↓). Class I and Class II MHC predicted to bind this epitope cover 22.14% and 12.69% of West Africa. It is not naturally processed in human APCs [49].	Faisal et al. [65] Boesen et al. [67]

**Table 4 viruses-12-00386-t004:** Advanced LASV vaccine candidates tested in “proof-of-concept” efficacy trials in non-human primates (NHPs).

Vaccine Candidate	LASV Vaccine Antigen(s)	Vaccine Regimen	Efficacy against LASV		Viremia after Challenge ^b^	Correlates of Protection	Ref
			Lineage IV ^a^	Other Lineages			
Recombinant vaccinia virus	GP (JOS) NP GP&NP	Single vaccination, at four sites, total 1 × 10^9^ PFU, ID	88% 20% 90%	ND	Low–moderate High Low–moderate	CMI	[47]
Reassortant MOPV/LASV, ML29	GP&NP (JOS)	One dose, 1 × 10^3^ PFU, SC	100%	II–100% (*guinea pigs*)	<LD	Sterilizing CMI	[21,69,70,88]
rVSVΔG/LASVGPC	GP (JOS)	One dose, 1–6 × 10^7^ PFU, IM	100%	I and V–100% (*guinea pigs*)	Low, transient	NAbs? CMI?	[23,46]
MV-LASV	GP & NP (JOS)	One dose, 2 × 10^6^ TCID_50_, SC	100%	ND	<LD	Non-NAbs and CMI	[51]
rVSV-N4ΔG-LASVGPC in Quadrivalent VesiculoVax	GP (JOS)	Two doses, 1 × 10^7^ PFU, IM	-	II–100%	<LD in 4 of 5 NHPs.	ND	[86]
VEEV-TC83 RNA replicon particles	GP (JOS&LP)	Two doses, 1 × 10^7^, SC	80%	20%	Moderate	ND	Lukashevich, Unpublished [27]
MOPEVAC_LASV_	GP (JOS)	One dose, 6 × 10^6^ PFU/dose, IM	100%	ND	Low, transient	Non-NAbs, CMI	[51]
DNA	GP (JOS)	Two immunizations, 20mg DNA at four sites, ID electroporation	100%	ND	ND	NAbs? CMI?	[78,79]

^a^ Challenge dose: 1 × 103–1 × 104 PFU of LASV/JOS (Lineage IV), route of inoculation: SC or IM. LASV-Z32 (Lineage IV) was also used [23]. ^b^ Low–moderate, 103–104 PFU/mL; high, >104 PFU/mL. Abbreviations: JOS, LASV Josiah strain; LP, LASV LP strain (Lineage I); CMI, cell-mediated immunity; FFU, fluorescent forming units; GP, glycoprotein; GP and NP, simultaneous expression of NP and GP in the same vector; ID, intradermal; IM, intramuscular; LASV, Lassa virus; LD, limit of detection; NAbs, neutralizing antibody responses; ND, not done; NHP, non-human primate; NP, nucleoprotein; PFU, plaque-forming unit; SC, subcutaneous. Modified from: Lukashevich, I.S.; Paessler, S.; de la Torre, J.C. Lassa virus diversity and feasibility for universal prophylactic vaccine. F1000Research 2019, 8, F1000 Faculty Rev-134.

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
