# Peer review of "Inter-Lineage Variation of Lassa Virus Glycoprotein Epitopes: A Challenge to Lassa Virus Vaccine Development"

_viruses, 2020, doi:10.3390/v12040386_

Round 1

Reviewer 1 Report

Type of manuscript: Review

Title: Inter-Lineage Variation of Lassa Virus Glycoprotein Epitopes: A Challenge to Lassa Virus Vaccine Development by Francis Ibukun

Journal: Viruses

This review targets LASV genetic diversity which is a serious challenge for development of LASV vaccines with “full coverage” as described in the WHO Target Product Profile (TPP) for LASV vaccine candidates. The review is well written. The author has demonstrated knowledge in the field. The review includes recently published data. Nevertheless, the interpretation of some of these results is confusing and misleading. The author is mostly focusing on LASV GPC variations since LASV GP1 and GP2 glycoproteins are responsible for the virus attachment, entry, induction of T cell responses, and interaction with neutralizing antibodies (nAbs). Several LASV vaccine candidates are based on expression of LASV GPC from Josiah strain (phylogenetic lineage IV). With great LASV genetic diversity and inter-lineage variation of GPC epitopes, the major question is if expression of LASV/Josiah GPC alone will be enough to induce a long-term sterilizing cross-protection against Nigerian and non-Nigerian strains of LASV. With all controversial data available so far, the author has to provide his view and, at least, partially answer this question. It will make this review helpful for vaccine developers and new comers in the field. Growing body of evidence indicates that anti-LASV NP responses are crucial to viral control at early stage of the infection and, perhaps, for long-term and broad cross-protective T cell immunity.

Comments:

Lines 38-40. “LASV is transmitted through oral or broken skin exposure to infected excreta of the multimammate rat species – Mastomys natalensis and M. erythroleucus – and Hylomyscus pamfi…” In fact, LASV major routes of transmission are respiratory and gastro-intestinal tracts. Persistently infected M.natalensis is the major natural source of rodent-to-human transmission. While the virus can be hosted by M. erythroleucus and H. pamfi, there is no documented epidemiological evidence of LASV transmission to human from these rodent species.

Lines 71-72. “LASV lineages differ in virulence properties: lineage IV infections have abundant viral genomes with a higher codon adaptation index than the Nigerian lineages [11]”. This statement is confused and misleading since viral RNA abundance does not directly correlate with pathogenicity. In contrast, high RNA/PFU ratio (e.g., due to presence of DI particles) can reduce lethality in non-human primates (Baize et al., 2009).

Lines 75-78. Susceptibility of guinea pigs with different genetic background to LASV strains from different phylogenetic groups is crucial for development of small animal challenge model. This information has to be supported by original references rather than by a review article.

Lines 97-98. “LASV diversity is reflected in its surface antigen the glycoprotein (GP)…”. In fact, LASV NP is the more diverse gene/protein than GP (Bowen et al., 2000).

Lines 129-130. “…adaptive immune protection in LASV infection is probably conferred through an interplay of dual cellular and humoral immune response that is dependent on early activation of 130 innate immune and inflammatory pathways…”   There is a strong body of evidence that T cell responses play major role in recovery and protection during natural LASV human infection. Humoral responses do not correlate with clinical manifestations and/or recovery. In fact, in most people, and in experimentally infected monkeys, these Abs are never detectable in a classical plaque-reduction neutralization assay (Fisher-Hoch & McCormick, 2001). Nevertheless, human mAbs generated in vitro from B cells of LASV-infected individuals, can be therapeutically active in treatment of LASV-infected guinea pigs and NHPs. 

Lines 132-133. “Non-neutralizing antibody (non-Nab) dependent ADCC/ADCP functions is the likely major purveyor of humoral immune protection [39,40], while neutralizing antibodies (Nab) arise late in …”. This speculation is based on experiments in animal models. There is no evidence that non-Nabs play a role in protection during natural LASV infection.

Lines 160-162. “In a study by Robinson et al, LASV GP B-cell epitopes were mapped using 113 human monoclonal antibodies (MAbs) derived from convalescent plasma of LF survivors, 15 from Sierra Leone and 2 from Nigeria [45].” This is a wrong and misleading statement, simply because mAbs cannot be derived from convalescent plasma. In the referenced publication, mAbs were generated in vitro from B cells of subjects with previous LASV exposure.   

Lines 282-284. “Furthermore, recent vaccine studies have also shown that Non-NAbs are important for protection against clinical disease in guinea pig and NHP models [39,40].” In fact, experiments in guinea pigs extensively immunized with inactivated experimental LASSARAB, rabies vaccine expressing LASV GPC, combined with adjuvant (TLR4 receptor agonist, GLA-SE), resulted in incomplete protection. Indeed, guinea pigs immunized twice showed no significant protection and 20% of surviving animals across all vaccinated groups had a very high RNAemia (105 LASV RNA copies/ml). With these results, it is hard to make a conclusion that Non-nAbs are important for protection since protection was very poor. In general, these experiments are in line with previously published findings. Extensive immunization of non-human primates with inactivated LASV resulted in Abs responses but did not protect animals against fatal LF (McCormick et al., 1992).

Lines 323-325. “Quaternary epitopes that combine multiple sites in different subunits of the GP complex are broadly cross-reactive across LASV lineages [26,45].” This is overstated and, perhaps, misleading conclusion because the cross-reactivity was measured in pseudo-virus neutralization assays. LASV/HIV-1 pseudo-types are artificially sensitive to neutralization and poor mimic infectious LASV neutralization (see Fig. 2a in Robinson et al., 2016).

Lines 369-370. “Sterilizing immunity – both cell-mediated and humoral – is probably essential to prevent immune-mediated neurologic sequelae of LF...”. There is no evidence that during natural LASV infection humoral responses induce sterilizing immunity. Likewise, there is no evidence of immune-mediated pathology in human LF. All available data are generated in animal studies, mostly in murine models.

Lines 381-383. “Sterilizing immunity, the “Holy Grail” of vaccine-conferred protection, may likely depend on LASV GP; while co-expression with LASV NP likely increases the breadth of protection and potency of the vaccine immunogen”. Vaccine-induced protection seems to be dependent on expression vector (e.g., rVSV-based vaccine predominantly induced humoral responses, while Ad-based vectors can drive both, humoral and T cell responses). The major contribution of NP is to induce effective viral control at early stage of the infection since in the absence of NP synthesis in vivo there are no detectable CD8+ T cell responses and protective immunity (Schildknecht A et al., 2008; Darbre S et al., 2015). In addition, in long-term survivals, anti-LASV NP T cell responses were broadly cross-reactive (ter Meulen J et al., 2000).

Author Response

Dear Reviewer 1,

I appreciate your comments and suggestions on this review article. I have addressed your comments by elaborating more on statements that were confusing due to over-summarization, corrected errors identified and incorporated your suggestions including recent evidence in the appropriate segments as well.

Comments:

Lines 38-40. “LASV is transmitted through oral or broken skin exposure to infected excreta of the multimammate rat species – Mastomys natalensis and M. erythroleucus – and Hylomyscus pamfi…” In fact, LASV major routes of transmission are respiratory and gastro-intestinal tracts. Persistently infected M.natalensis is the major natural source of rodent-to-human transmission. While the virus can be hosted by M. erythroleucus and H. pamfi, there is no documented epidemiological evidence of LASV transmission to human from these rodent species.

Response 1: This has been elaborated upon based on your suggestion in Lines 38 – 43: “LASV is mainly transmitted as a zoonosis, by ingestion of food and inhalation of aerosols contaminated with infected excreta of the multimammate rat species – Mastomys natalensis – which serve as a reservoir for LASV [9]. Broken skin exposure to infected rodent blood is also acknowledged as a risk factor for LF [10]. While LASV has been found in M. erythroleucus and Hylomyscus pamfi [11], there is no documented epidemiologic evidence of LASV transmission to humans from these species.”

Lines 71-72. “LASV lineages differ in virulence properties: lineage IV infections have abundant viral genomes with a higher codon adaptation index than the Nigerian lineages [11]”. This statement is confused and misleading since viral RNA abundance does not directly correlate with pathogenicity. In contrast, high RNA/PFU ratio (e.g., due to presence of DI particles) can reduce lethality in non-human primates (Baize et al., 2009).

Response 2: While I understand your comment, these inter-lineage differences in LASV viral genome abundance and codon adaptation index need to be highlighted. I have revised the statement to show this as additional differences in LASV lineages, rather than specific virulence properties. A caveat about multifactorial contribution to observed CFR also follows. Lines 84-89: “Lineage IV LASV human infections also have more abundant viral genomes (a correlate of viral load) and a higher codon adaptation index compared to Nigerian lineages [12]. However, these have not been shown to cause the higher LF CFR seen in the MRU; other factors such as delay in seeking care, economy, poor regional healthcare system and host genetic factors also contribute to the regional difference in CFR [12].”

Lines 75-78. Susceptibility of guinea pigs with different genetic background to LASV strains from different phylogenetic groups is crucial for development of small animal challenge model. This information has to be supported by original references rather than by a review article.

Response 3: Original references have been added as recommended in Lines 74 – 78.

Lines 97-98. “LASV diversity is reflected in its surface antigen the glycoprotein (GP)…”. In fact, LASV NP is the more diverse gene/protein than GP (Bowen et al., 2000).

Response 4: The statement has been revised to include higher diversity in LASV NP, while retaining the focus of this article on LASV GP diversity. Lines 101-103: “LASV diversity is reflected in its surface antigen, the glycoprotein (GP) (Figure 2), and to a larger extent in its nucleoprotein (NP) [14]. This GP is commonly utilized as immunogen in vaccine development, with the addition of NP in fewer candidates.”

Lines 129-130. “…adaptive immune protection in LASV infection is probably conferred through an interplay of dual cellular and humoral immune response that is dependent on early activation of innate immune and inflammatory pathways…”   There is a strong body of evidence that T cell responses play major role in recovery and protection during natural LASV human infection. Humoral responses do not correlate with clinical manifestations and/or recovery. In fact, in most people, and in experimentally infected monkeys, these Abs are never detectable in a classical plaque-reduction neutralization assay (Fisher-Hoch & McCormick, 2001). Nevertheless, human mAbs generated in vitro from B cells of LASV-infected individuals, can be therapeutically active in treatment of LASV-infected guinea pigs and NHPs. 

Response 5: I agree with your comments on this statement, on the major role of cellular immune response in recovery and protection in Lassa fever. I have revised this statement to remove the conflation with the emerging role of humoral response seen specifically in LASV vaccine studies in animal models and not in natural human LASV infection. Lines 134 -136: “adaptive immune protection in LASV infection is probably conferred mainly by a cell-mediated immune response that is dependent on early activation of innate immune and inflammatory pathways – especially Type I IFN response [43–49].

Lines 132-133. “Non-neutralizing antibody (non-Nab) dependent ADCC/ADCP functions is the likely major purveyor of humoral immune protection [39,40], while neutralizing antibodies (Nab) arise late in …”. This speculation is based on experiments in animal models. There is no evidence that non-Nabs play a role in protection during natural LASV infection.

Response 6: I have elaborated on this speculation based on observations from LASV vaccine studies in animal models and the lack of evidence in human Lassa fever. Lines 136 - 139: “While it is speculated that non-neutralizing antibody (non-Nab) dependent ADCC/ADCP functions is the likely purveyor of humoral immune protection, as seen in LASV vaccine studies in animal models [50,51]; there is no evidence for this role in human LF cases. Whereas, neutralizing antibodies (Nab) arise late in natural infection…”

Lines 160-162. “In a study by Robinson et al, LASV GP B-cell epitopes were mapped using 113 human monoclonal antibodies (MAbs) derived from convalescent plasma of LF survivors, 15 from Sierra Leone and 2 from Nigeria [45].” This is a wrong and misleading statement, simply because mAbs cannot be derived from convalescent plasma. In the referenced publication, mAbs were generated in vitro from B cells of subjects with previous LASV exposure.   

Response 7: I have corrected this statement in Lines 166 -168: “In a study by Robinson et al, LASV GP B-cell epitopes were mapped using 113 human monoclonal antibodies (MAbs) derived in vitro from blood of LF survivors, 15 from Sierra Leone and 2 from Nigeria.”

Lines 282-284. “Furthermore, recent vaccine studies have also shown that Non-NAbs are important for protection against clinical disease in guinea pig and NHP models [39,40].” In fact, experiments in guinea pigs extensively immunized with inactivated experimental LASSARAB, rabies vaccine expressing LASV GPC, combined with adjuvant (TLR4 receptor agonist, GLA-SE), resulted in incomplete protection. Indeed, guinea pigs immunized twice showed no significant protection and 20% of surviving animals across all vaccinated groups had a very high RNAemia (105 LASV RNA copies/ml). With these results, it is hard to make a conclusion that Non-nAbs are important for protection since protection was very poor. In general, these experiments are in line with previously published findings. Extensive immunization of non-human primates with inactivated LASV resulted in Abs responses but did not protect animals against fatal LF (McCormick et al., 1992).

Response 8: I agree with your comments about the shortcomings of the inactivated LASSARAB vaccine study. However, two recent LASV vaccines studies – the live attenuated Measles virus vectored LASV vaccine tested in NHPs (Mateo M et al, 2019) and the live attenuated LASV with codon deoptimized  GP vaccine tested in guinea pigs (Cai Y et al 2020) – have also highlighted the correlation of non-neutralizing antibodies (Non-Nabs) to vaccine induced protection after lethal LASV challenge in animal models. As such I have revised the statement above, to highlight (but not overstate) the contribution of Non-NAbs to vaccine-induced protection. Lines 288–290: “Furthermore, recent vaccine studies have also shown that Non-NAbs may contribute to protection against clinical disease in guinea pig and NHP models [50,51,62].

Lines 323-325. “Quaternary epitopes that combine multiple sites in different subunits of the GP complex are broadly cross-reactive across LASV lineages [26,45].” This is overstated and, perhaps, misleading conclusion because the cross-reactivity was measured in pseudo-virus neutralization assays. LASV/HIV-1 pseudo-types are artificially sensitive to neutralization and poor mimic infectious LASV neutralization (see Fig. 2a in Robinson et al., 2016).

Response 9: I agree with your comments and I highlighted this shortcoming in an earlier section (Lines 177-182). I have now added this caveat to the statement in Lines 335-339: “Quaternary epitopes that combine multiple sites in different subunits of the GP complex are broadly cross-reactive across LASV lineages [32,53]; however, the cross-reactivity was measured in LASV/HIV-1 pseudo-virus neutralization assays which are artificially sensitive to neutralization and poor mimic of authentic LASV PRNT. GP2 epitopes also appear well conserved.

Lines 369-370. “Sterilizing immunity – both cell-mediated and humoral – is probably essential to prevent immune-mediated neurologic sequelae of LF...”. There is no evidence that during natural LASV infection humoral responses induce sterilizing immunity. Likewise, there is no evidence of immune-mediated pathology in human LF. All available data are generated in animal studies, mostly in murine models.

Response 10: I agree with your comments on the evidence for immune-mediated sequelae being limited in animal models. Vaccine-induced protection from these immune-mediated sequelae was also demonstrated in animal models. Per your comment on the absence of evidence for sterilizing humoral immunity and immune-mediated pathology in natural human LASV infection, I have revised this statement in segment discussing the role of NP in inducing cell-mediated immune response while incorporating your comments on NP below. Lines 400-403: “It is also speculated to prevent immune-mediated neurologic sequelae of LASV infection in animal models, such as sensorineural deafness, that is due to persistent viremia in immunologically privileged tissues [51,76–78]; however, there is no evidence of immune-mediated pathology in human LF so far.”

Lines 381-383. “Sterilizing immunity, the “Holy Grail” of vaccine-conferred protection, may likely depend on LASV GP; while co-expression with LASV NP likely increases the breadth of protection and potency of the vaccine immunogen”. Vaccine-induced protection seems to be dependent on expression vector (e.g., rVSV-based vaccine predominantly induced humoral responses, while Ad-based vectors can drive both, humoral and T cell responses). The major contribution of NP is to induce effective viral control at early stage of the infection since in the absence of NP synthesis in vivo there are no detectable CD8+ T cell responses and protective immunity (Schildknecht A et al., 2008; Darbre S et al., 2015). In addition, in long-term survivals, anti-LASV NP T cell responses were broadly cross-reactive (ter Meulen J et al., 2000).

Response 11: I have revised this statement to elaborate on the role of NP in vaccines that co-expresses both LASV GP and NP. I also incorporated your comments and evidence from a recent study of cross-reactive anti-LASV T-cell responses (Sullivan et al, 2020) to this section as well. Lines 388-391: “In human LF survivors, CD4+ T cell responses were mainly directed towards LASV NP, and anti-LASV NP CD4+ T-cell responses were broadly cross-reactive, while cross-reactive anti-LASV NP CD8+ T-cell responses target relatively conserved NP epitopes [66,71].” Lines 395-399: “Even though, vaccine-induced immunity appears dependent on expression vector, the co-expression of LASV GP and NP may also contribute to induction of sterilizing immunity – the “Holy Grail” of vaccine-conferred protection. NP induces effective viral control at the early stage of arenavirus infection; since in the absence of NP synthesis in vivo, there are no detectable CD8+ T-cell responses and protective immunity [74,75].”  

I have also given my view on the induction broad cross-protection by a vaccine expressing only LASV/Josiah GPC in lines 493-496: ”Given the importance of cell-mediated immune protection in LF that is relatively biased towards LASV NP and evidence of cross-reactive T-cell responses in LF survivors, it seems that vaccines co-expressing LASV GP and NP of a single lineage (Lineage IV/Josiah strain) will more likely induce broader cross-protection against other LASV lineages with a potential for sterilizing immunity.”

Reviewer 2 Report

Lassa fever is a severe and fatal zoonotic disease that is endemic in Western Africa. Vaccination is arguably the most cost-effective countermeasure against the disease in endemic areas. Lassa virus, the causative agent of Lassa fever, is highly divergent in the genomic sequence among different isolates/strains. Most of the vaccine studies use the Josiah strain to test the efficacy of vaccine candidates. Considering the high variation in viral genomic RNA sequences, which could be as high as 25% in S segment, a pan-LASV vaccine is desirable. This paper by Francis Ibukun nicely reviewed the inter-lineage variations within previously mapped B-cell and T-cell LASV GP epitopes. Additionally, LASV vaccine/challenge studies were examined critically. The manuscript is well written and provides comprehensive and in-depth review of the topic, which is useful for researchers in the field.

Minor question:

Line 179: the cited ref used a guinea pig adapted LASV (Josiah) in animal infections. Is there any information regarding the viral sequence changes after adaptation?

Author Response

Dear Reviewer 2,

I appreciate your comments on this review article and I have responded to the minor question asked below. Thank you for your time and consideration.

Minor question:

Line 179: the cited ref used a guinea pig adapted LASV (Josiah) in animal infections. Is there any information regarding the viral sequence changes after adaptation?

Response 1: The viral sequence for the L and S segments of the guinea pig adapted LASV (Josiah) is available in the Genbank: (stock IRF0205; L segment GenBank KY425651.1; S segment GenBank KY425643.1) [1]. Its L and S segments have 99.88% and 99.97% nucleotide sequence identities with the wild type LASV Josiah virus. Additionally, there are no mutations within the coding region (CDS) of the four genes (Z, L, NP and GPC).

  1. Abreu-Mota, T.; Hagen, K.R.; Cooper, K.; Jahrling, P.B.; Tan, G.; Wirblich, C.; Johnson, R.F.; Schnell, M.J. Non-neutralizing antibodies elicited by recombinant Lassa-Rabies vaccine are critical for protection against Lassa fever. Nat. Commun. 2018, 9, 4223–4223.

Round 2

Reviewer 1 Report

The revised manuscript has been significantly improved. The author addressed all questions raised by the reviewer. 

Two minor issues have to be fixed:

Line 137-139, "...Nab does not correlate with protection..." In most cases,  protection" is a feature assessed in challenge experiments in animal models. In cases of human natural LASV infections, we are dealing with clinical manifestations and final outcome, death or recovery. It is more correctly and more important to emphasize that improvement in clinical manifestations and recovery of natural LASV infections in humans DO NOT correlate with humoral immunity and Nab appeared too late to make any contributions to clinical recovery and/or outcome. 

Line 146, it is not clear what does it mean "GP potency". In vaccinology, potency is a major biological feature of replication-competent vaccines (in PFU or in infectious units). The author has to specify GP potency definition.

Author Response

Dear Reviewer 1,

Thank for taking the time to review this article. I have addressed the minor issues below.

Two minor issues to be fixed:

Line 137-139, "...Nab does not correlate with protection..." In most cases,  protection" is a feature assessed in challenge experiments in animal models. In cases of human natural LASV infections, we are dealing with clinical manifestations and final outcome, death or recovery. It is more correctly and more important to emphasize that improvement in clinical manifestations and recovery of natural LASV infections in humans DO NOT correlate with humoral immunity and Nab appeared too late to make any contributions to clinical recovery and/or outcome.

Response 1: I have revised this statement to reflect the lack of contribution of NAbs to clinical recovery from Lassa fever and distinguished this from the evidence garnered from therapeutic neutralizing MAb study in animal models. Lines 139-142: “…neutralizing antibodies (Nab) arise too late in natural human LASV infection to contribute to clinical recovery [36]. Moreover, the in-vitro neutralizing activity of anti-LASV human monoclonal antibodies does not correlate with protection in animal models [45,52].”

Line 146, it is not clear what does it mean "GP potency". In vaccinology, potency is a major biological feature of replication-competent vaccines (in PFU or in infectious units). The author has to specify GP potency definition.

Response 2: I agree that vaccine potency is a major biological feature of replication-competent vaccines. “GP potency” here refers to the potency of vaccines expressing LASV GP. I have revised this statement to clearly reflect what I mean in Lines 149-151: “The GP is also regarded as the primary immunogen for LASV vaccine development [36], because vaccines expressing LASV GP conferred a higher level of protection than those expressing LASV NP only [23,47].”